Monitoring and analysis of the expansion of the Ajmr Port, Davao City, Philippines using multi-source remote sensing data

Li Humei 1
Wu Mingquan wumq@aircas.ac.cn 2
Tian Dinghui 1
Wu Lianxi 1 3
Niu Zheng 2
1 Faculty of Geomatics, East China University of Technology , Nanchang , Jiangxi , China
2 The State Key Laboratory of Remote Sensing Science, Institute of Remote Sensing and Digital Earth, Chinese Academy of Sciences , Beijing , China
3 Jiangxi Key Laboratory for Digital Land , Nanchang , Jiangxi , China
Yu Le
Electronic publication date: 2019 Aug 19
Publication date: 2019
Volume: 7
Electronic Location ID: e7512
Received 2019 May 6; Accepted 2019 Jul 18
Copyright: ©2019 Li et al.
Copyright year: 2019
Copyright holder: Li et al.
License: This is an open access article distributed under the terms of the Creative Commons Attribution License, which permits unrestricted use, distribution, reproduction and adaptation in any medium and for any purpose provided that it is properly attributed. For attribution, the original author(s), title, publication source (PeerJ) and either DOI or URL of the article must be cited.
License URL: https://creativecommons.org/licenses/by/4.0/

Keywords: Ecological environment, Small ports, Monitoring, Remote sensing, Philippines

Funding: Strategic Priority Research Program of Chinese Academy of Sciences XDA19030404 Youth Innovation Promotion Association CAS 2017089 This work was supported by the Strategic Priority Research Program of Chinese Academy of Sciences (No. XDA19030404) and the Youth Innovation Promotion Association CAS (No. 2017089). The funders had no role in study design, data collection and analysis, decision to publish, or preparation of the manuscript.

==============================
Ports have been built or expanded in a number of countries to cater to increasing maritime trade in the 21st century. Port expansion is associated with economic and environmental impacts on the local and regional scales, and these impacts can be studied using remote sensing. The present study presents new results from multi-source remote sensing monitoring of the Ajmr Port expansion. An analysis of land use and vegetation coverage at the port is used to monitor the impact of port construction on the local ecology, while changes in roads, buildings, and lights are used to monitor the economic impact. The results show that: (1) After nine years of expansion, the port area has gradually expanded from the central to the southern coastal area, with an increase of 21.68 hectares during the expansion period. After the expansion, the area of builidings and construction in the study area increased significantly, while the area of water and green areas decreased significantly, indicating that the port construction changed the land use structure of the area. (2) From the perspective of vegetation coverage, the vegetation coverage within 5 km from the port is in good condition. After 9 years, the vegetation coverage in the region between 0.6 and 1 increased from 43.71% to 44.25%, reflecting the higher overall greening level in the region. (3) By analyzing the increase in roads and buildings, it can be seen that the port’s comprehensive transportation capacity has improved, the population of the region has increased significantly. As the scale of construction has been continuously expanded , the prosperity as increased. (4) By analyzing the changes in the light index, the light data from the northeast to the southwest in the region is very obvious, and it is clearly located along the coast, indicating that the economic development of the coastal zone is faster than other regions, and the coastal region has promoted the development of the inland region.

Introduction

The “21st Century Maritime Silk Road” (MSR) is a major China-based initiative focused on domestic development, with benefits for international development throughout the region. Ports are a key component of the strategic infrastructure that supports the MSR transportation corridor. Additionally, ports will attract new resources into areas, and become important nodes that will promote new opportunities for the country and strengthen international trade. Port development will increase gross domestic product, improve national income, provide more employment, and it plays an important role in regional and urban economic development. However, construction of infrastructure at the port has had profound impacts on regional land use. These impacts include ecological and environmental problems, such as erosion in the coastal zone and pollution and disturbance of the marine environment (Gupta, Gupta & Patil, 2005; Peng et al., 2014). Increases in maritime trade have driven an increase in the number of ports internationally, and remote sensing has been used to monitor and analyze the impacts of port construction on the regional ecology. Previous work has focused on the effects that the expansion of large and medium-sized ports have had on the temporal and spatial evolution of the natural environment. However, the number of small port expansions is increasing rapidly, so the impact of these smaller developments must also be investigated.

The methods used for port monitoring, and the results, are diverse. For example, Cowell & Monk (1981) used measurements of the abundance of intertidal species near a dock in Valdez Port, Alaska, USA, to develop ecological for a rock bank baselines in the area. Gulbin, Arzamastsev & Shulkin (2003) used a survey of benthic communities in Wrangel Bay, Peter Dam, Sea of Japan, to reveal the environmental impacts of port operations in the area. A three-stage SPOT image was used to study changes in land use in the vicinity of the lower reaches of the Thi Vai River in Ho Chi Minh City and Vung Tau City, southern Vietnam (Rutten, Binh & Hens, 2005). Smith et al. (2007) used baseline data that represented water quality, sediment quality, and benthic diversity to assess the ecological status of coastal resources in the Gulf of Mexico. Chen & Luo (2009) used a combination of the comprehensive index method and the decision tree classification method to study changes in land used for construction in Qinzhou Port over an eight-year period. Kidd (2010) explored the relationship between the historic public policy of Houston, Texas, and the development and expansion of the Houston port system and the impact of port expansion on the socioeconomic status of surrounding communities. Portman, Jin & Thunberg (2011) used parcel-level data and geographic information system (GIS) tools to study the interactions between fisheries and changing land use for two New England ports. Zhu et al. (2012) used a comprehensive analysis of land use in the coastal areas surrounding Bohai Bay to study the evolution of land use dynamics and spatial differentiation. Liu et al. (2012) established four land use change index models, and used them to analyze the orientation and extent of spatial–temporal patterns and expansion patterns associated with changes in land use type in and around the Tianjin Port Area between 1987 and 2010. Chen used remote sensing data to simulate the process of changing land use for the port cities of Dalian, China. The results were used to support decisions that promoted sustainable development in port cities (Chen & Nuo, 2013). Bezerra et al. (2014) discussed the economic development of the Port of Pecém and Industrial Complex in Ceara, Brazil, and its impact on employment, environment, and health. Felsenstein, Lichter & Ashbel (2014) used microscopic simulation models and statistical estimates to determine the relationship between expansion of the Haifa Port and expected future land use. Enaruvbe & Ige-Olumide (2014) used remote sensing and GIS to analyze the patterns and processes of changing land use around Port Harcourt, which is Nigeria’s second largest port. These researchers monitored an increased rate of conversion from mangroves, wetlands, and water bodies to cultivated land, and a reduced rate of growth of the area of secondary vegetation and natural forests. Liu (2014) used the normal cloud model to establish a comprehensive set of ecological indices for ports, and evaluated the ten largest Chinese ports, providing a basis for future port ecological development. Sany et al. (2015) developed biological indicators based on large benthic communities, and used the indicators to assess the ecological status of the marine environment. Zhao & Liu (2015) used the quantitative method of R clustering and coefficient of variation, combined with the qualitative method of expert experience, to study the 17 major ports such as Shanghai Port and Ningbo-Zhoushan Port, that proved the rationality of the ecological port evaluation index system. Li et al. (2016) established a more detailed and applicable classification system, using multi-source and multi-resolution images to monitor and analyze land use changes in and around three typical ports in the vicinity of Xiamen Port. Hassan & Omran (2017) used remote sensing and GIS to monitor changes in land use and land cover in the southern part of the province that hosts Port Said in Egypt, and to assess the impact of the port on climate change. Huang & Liao (2017) used dynamic monitoring of remote sensing images and long-term observations to determine variation range at important ports in the Beibu Gulf. The results were combined with other data, including statistical data, to produce an analysis of the development of port construction and its impact on parameters such as terminal berths and cargo throughput. Ge et al. (2018) used multi-temporal high spatial resolution remote sensing images to monitor infrastructure construction at the ports of Colombo and Hambantota between 2010 and 2017; the results were used to track the progress of the port construction, and to predict increases in annual port throughput through potential water storage. Li, Xu & Su (2019) used the five typical ports in the Bohai Sea as an example to establish a remote sensing monitoring technology process for the port space pattern, and constructed an index of various objects to quantitatively describe the port space composition and performance. Zhong et al. (2019) constructed a mangrove wetland ecosystem health evaluation index based on Google Earth images, and explored the spatial and temporal differentiation of the health of the mangrove wetland ecosystem in the Zhangjiang Estuary, and the factors that influenced the wetlands.

These previous studies have focused on the ecological and economic aspects of ports or coastal areas, including ecological aspects such as water quality, sediment quality, species diversity, cultivated land change, vegetation cover, mangrove status, wetland changes, and land use. Economic aspects considered in the studies include rates of construction progress, construction land use, level of industrial development, number of berths, port throughput, and public policies. Research has generally focused on ecological or economic aspects; few remote sensing monitoring analyses have focused on both. To address this, we use a comprehensive analysis of remote sensing images to investigate ecological and economic consequences of the expansion of Ajmr Port, Davao City, in the Philippines.

The ecological environment is studied from the land. As a natural economic complex, land is not only an important part of the human-land system, but also the main space support for ecological civilization construction. Land use change is a core area of global environmental change and sustainable development research, affecting and changing the structure and function of the eco-economic system (Yu et al., 2010). In addition, we study ecology from the perspective of vegetation. As an important part of the ecosystem, vegetation has an important impact on the energy flow, material circulation and information transmission of surface ecosystems. Vegetation coverage can reflect the state and structure of vegetation growth and is an important indicator for climate change, soil erosion and hydrological processes. By monitoring land use change and vegetation coverage, we can understand the impact of port construction on the ecological environment of the area.

There are many indicators for social and economic development, such as roads, buildings, population, GDP, and per capita consumption levels. This paper selects three indicators for monitoring and analysis. The first indicator is the road, which is an important factor in the development of the regional economy. The radiation situation of road traffic reflects the radiation situation of the regional economy to a certain extent. The accessibility and convenience of the road have greatly promoted the development of the economy. The second indicator is building area, which is an important technical and economic indicator. In a certain period of the national economy, the completion of the construction area also marks the development of industrial and agricultural production in a country, the improvement of people’s living and living conditions and the development of cultural and welfare facilities. The third indicator is the light index, which was first proposed by American scholars to point out that the brightness of a region’s night is proportional to its GDP. That is, the higher the light index value and the greater the change, the higher the prosperity of the representative city and the faster the economic development. By monitoring road changes, building area changes and light indices in the area, the economic impact of port construction on the area can be reflected.

Study Area and Data Processing

Study area

Ajmr Port is located in the northern part of Davao City, Philippines, 15 km from Davao City. The city of Davao is located on the land about 946 km southeast of Manila (Fig. 1). The city covers a total area of 2,443.61 square kilometers and is the largest city in the Philippines and the third most populous metropolitan area in the Philippines.There are several corporate terminals along the coast of Davao City, including Texaco, Legazpi, National Warehouse, and the Philippines Shell Oil Terminal.

Figure 1 Research area location.

©2018 Google Earth.

Ajmr Port is mainly used to import and export goods for the Sumifru Company, so it is also known as Sumifru Port and is part of the Philippines commercial port system. The Sumifru Company is a large-scale international enterprise that integrates planting, research and development, and import and export of tropical fruit products, including pineapple, banana, and mango, with banana production accounting for three-quarters of the company’s total fruit cultivation. Davao has a tropical rainforest climate with little seasonal variation in temperature. Thus, the monthly average temperature is always relatively high with an average monthly rainfall of over 77 mm. The country’s bananas are abundant in the mountains at elevations above 600 m. The Sumifru Company currently has 12,010 hectares of banana plantations in five different regions of Davao City, Philippines. The total output and export volume of bananas ranks second among the Philippines-related industries, and its fruits are exported to China, Japan, South Korea, Iran and many other countries.

Image data and pre-processing

High-definition images that record the early stages of port construction are available from 2008, 2009, 2010, and 2012. This paper describes the process of the expansion of Ajmr Port. The high-definition images show that the port of Ajmr changed significantly in 2012, indicating that the port began to expand after 2010. Images from 2009 to 2018 were selected for analysis (Fig. 2).

Figure 2 High-definition images of the 2008 (A), 2009 (B), 2010 (C) and 2012 (D) in study areas.

(A) ©2008 Google Earth, (B) ©2009 Google Earth, (C) ©2010 Google Earth, (D) ©2012 Google Earth.

Landsat images of the port taken between 2009 and 2018 were selected to monitor the vegetation cover before and after the port expansion. Images from Landsat7 and Landsat8 were downloaded for 2009 and 2018, respectively. However, as a result of the failure of a Landsat 7 sensor in 2003, a large number of bands appeared in the image. These artefacts were repaired using the Landsat_gapfilter.sav plug-in, and by pre-treating the images in ENVI; processing included geometric corrections, image clipping, and atmospheric and radiation corrections. A square image centered on the port, with a side length of five kilometers, was obtained after the correction.

High-definition images were selected for the port area from 2009 to 2018 and used to classify the land use. The expansion lasted for nine years, and changes were continuous throughout this period. Monitoring for the present study, we used images taken at three-year intervals, with images from 2009, 2012, 2015, and 2018. These high-resolution images were obtained from Google Earth; resolution is 1 meter. The images were corrected using the ENVI software so that the sample points were evenly distributed throughout the image, and the images were registered using the image to map method. The corrected images are amenable to dynamic change analysis.

Time-series nighttime images from 2009 to 2018 were downloaded. Processing of the lighting images required extensive observation of the surrounding areas of the port, so the region of interest used for the study of night light data was a 40 km ×40 km, centered on the port. The night lighting data from 2009 to 2013 was taken from DMSP/OLS (operational line scanning system of the Defense Meteorological Satellite Program) and from VIIRS/DNB (visible infrared imaging radiometer kit day/night band) for 2013 to 2018. Compared with the OLS data, the VIIRS benefits from a finer space–time resolution, better data comparability between different years, and no data saturation. The acquisition of data from two different observation platforms using multiple sensors means that the downloaded images have to be pre-processed (Zou et al., 2014; Shi et al., 2014; Cao et al., 2015; Zhang, He & Fan, 2017; Wu, Li & Zhang, 2018). First, the images were cropped so they covered the same area. Then, the image was re-sampled so that all images had the same spatial resolution. A quadratic regression model was established to correct for differences between the sensors. (1) DNc=a×DN2+b×DN+c,

where DN and DNC are the grayscale values before and after correction, and a, b, and c are regression parameters. Then, the DN values of the two sensor stable bright value pixel images in the same year are corrected. (2) DNn,i_c=DNn,iα+DNn,iβ2,

where DNn,iα and DNn,iβare the uncorrected DN values of pixel i acquired by sensors α and β for year n, and DN(n,i)_c is the corrected DN value of pixel i in year n. Finally, the DN value correction of the multi-sensor multi-year bright value pixel image is performed according to the change rule of the night light image. In cases where the DN value of a pixel in the DMSP/OLS light image for a particular year was less than the DN value for that pixel in the previous year, then the DN value should be corrected. (3) DNn,i=DNn−1,i,DNn−1,i>DNn,iDNn,i,others,

where DN(n−1,i) and DN(n,i) represent the DN values of pixel i on the DMSP/OLS nighttime light image in years n − 1 and n, respectively.

Methods

The Google Earth High Definition images were used to produce maps of the type of land use and to establish the timing of changes in land use. The Landsat images were used to create vegetation coverage maps and processed to assess changes in vegetation growth during port expansion; these data were used to monitor the ecological environment. The Google Earth High Definition images were used to monitor the progress of port roads and building construction, and measure the density of port transportation infrastructure and housing density. These data were combined with the light images to calculate the annual brightness value of the port, plot the rate of change of light around the port, and to analyze the impact of small port construction on the local economy.

Port ecological environment remote sensing monitoring analysis

Land use monitoring

After pre-processing of the 2009, 2012, 2015, and 2018 high-resolution images from Ajmr Port, visual interpretation was used to analyze spatio-temporal dynamic changes. The scheme for the port classified features as roads, buildings, water, pier, container yard, bare land, other facilities, green areas, residential areas, or construction areas. The roads were classified as dirt roads, cement roads, or internal roads, and the area where the port is located was marked. The criteria used to classify the interpreted features are provided (Table 1 & Fig. 3).

Table 1 Local object identification and judgment methods.

Feature type	Color, shape and obvious features	
Road	Dirt road: the color is grayish white, the road is thinner and the shape is irregular (Fig. 3A); the cement road: the color is grayish black, the road is straight and very wide (Fig. 3B); roads within the port: the color is gray, long Strips, wide roads, located in the inner area of the port (Fig. 3C)	
Building	The colors are white, dark green, blue, etc., the shape is a regular polygon, the area is large, and there are generally roads around (Fig. 3D)	
Waters	The color is dark green, which is a large area, and there are a small amount of sandstone and silt accumulation in the coastal area (Fig. 3E).	
Pier	The color is gray, and the shape is a regular rectangle. In the coastal area, there are ships nearby, and there are generally goods and vehicles on the dock (Fig. 3F).	
Container yard	The colors of the containers are different, the shape of the yard is relatively regular, and there are many roads and containers inside (Fig. 3G).	
Bare land	The general color is brown, the shape is irregular, and there is no vegetation, construction and construction facilities inside (Fig. 3H)	
Other facilities	The color is dark blue or dark gray, and the area is relatively regular, generally for various mechanical equipment or workshops (Fig. 3I)	
Green space	The color is green, it is a flaky area, and there are various trees, grasses and other vegetation inside (Fig. 3J)	
Residential area	The color is mostly red and blue, and the flaky area is irregular. There are roads and a small number of trees inside (Fig. 3K)	
Construction area	The color is grayish white, irregular, with obvious construction traces or construction equipment (Fig. 3L)	

Figure 3 Visual interpretation of various feature.

(A) dirt road, (B) cement road, (C) roads within the port, (D) building, (E) waters, (F) pier, (G) container yard, (H) bare land, (I) other facilities, (G) green space, (K) residential area, (L) construction area (A–F): ©2015 Google Earth, (G–L): ©2016 Google Earth.

The 2009 image was interpreted first. The interpreted image was compared with the next image (2012), and modified to create the 2012 land use map. The land use maps for 2015 and 2018 were derived in an analogous manner. Changes in the land use related to construction were determined by comparing the images.

Vegetation growth sequence monitoring

The normalized difference vegetation index (NDVI) for 2009 was calculated, based on the 2009 pre-processed Landsat imagery, and used to estimate the fractional vegetation cover (VFC) in the Ajmr Port region. First, the NDVI of the Landsat remote sensing image in 2009 was calculated, and then the maximum NDVI of the year was synthesized. Based on this image, the VFC was calculated using empirical equations. (4) NDV I=Pnir−Pred∕Pnir+Pred,

where Pnir isthe reflectance of the near-infrared band and Pred is the reflectance of the red band, and (5) V FC=NDV I−NDV Imin∕NDV Imax−NDV Imin

where NDVImax and NDVImin are the maximum and minimum NDVI values in the region, respectively.

To account for the inevitable noise in the images, NDVImax was set to the 95th percentile of the DN data, and NDVImin was set to the 5th percentile of the DN data; this strategy minimized noise-related artefacts in the image. The same approach was used for the 2018 vegetation cover map. A comparison of the two images was used to determine changes in vegetation cover before and after port construction.

Social and economic remote sensing monitoring analysis

Comparison changes in roads and buildings

Roads are an important factor in the development of a regional economy, and the volume of road traffic reflects the regional economy to some extent. The accessibility and convenience of roads promotes economic development (Wang, 2014; Li, 2014; Yuan, 2018). In this study, changes to the roads inside the port region were determined, and divided into three grades according to the road construction status: dirt, cement, and the roads within the port. The roads were extracted from the high-resolution images. A comparison of images taken before and after the expansion project enabled visualization of traffic changes in the area, the density of the road network, the estimated traffic volume, and the extent of development of the port city.

The economic prosperity of the area can be estimated using a comparison of buildings in the port area before and after port construction. The number of buildings was counted, and the data used to calculate the area occupied by buildings and visualize the progress of port construction.

Analysis of light changes

Nighttime light is an objective measure of the productivity and activity of human society (Dolores, 2013; Xu et al., 2015; Wang & Huang, 2018; Li et al., 2018). The lighting data were used to study development and activity in the port area, using a comparison of the bright and dark regions in the images. There is a positive correlation between the intensity of lighting and socio-economic factors. This paper used nighttime lighting data from 2009 to 2018 to determine changes to lighting in the port area. Comparisons of the lighting data over the nine-year period were used to identify areas where light changed in the study area, and to infer changes in demographic and the status of economic development.

Results and analysis

The status of land use in the study area

It was not possible to conduct field verification of features measured in the older images, because of changes to the port in recent years. Instead, the high-precision 0.3-m-resolution high-definition image was used to verify the accuracy and revise the results where necessary. The accuracy of the corrected data is better than 90%, which is considered to be satisfactory for the purposes of the study. At the time of construction, the area covered by the port was 9.46 hectares. The port expanded to 19.96 hectares in 2012, 26.95 hectares in 2015, and 31.14 hectares in 2018 (Fig. 4).

Figure 4 Port expansion chart of the 2009 (A), 2012 (B), 2015 (C) and 2018 (D).

(A) ©2009 Google Earth, (B) ©2012 Google Earth, (C) ©2015 Google Earth, (D) ©2018 Google Earth.

Maps of land use type for 2009, 2012, 2015, and 2018 were constructed (Fig. 5). There were large changes in land use type in the study area during port construction. The area occupied by the port increased by a total of 21.68 hectares during the expansion period, mainly through the addition of land previously occupied by water. From 2009 to 2018, port expansion gradually shifted from the central to the southern coastal region, mainly at the expense of green space. The most obvious features that changed during the expansion period are the areas of buildings, construction, bare land, water, other facilities, container yards, roads, and green space. Among these, there was a significant increase in the area occupied by buildings, construction, bare land, other facilities, container yards, and roads, with increases of 10.05, 8.58, 5.1, 4.43, 8.68 and 4.06 hectares, respectively. These increases were accommodated by a reduction in area occupied by water and green space, with reductions of 12.02 and 29.95 hectares, respectively.

Figure 5 Land use change map of the study area in 2009 (A), 2012 (B), 2015 (C) and 2018 (D).

Table 2 Area of various types of features in the study area (hectare).

Feature category	August 2009	August 2012	July 2015	July 2018	
Building	6.87	9.37	12.21	16.92	
Waters	118.53	112.55	109.28	106.51	
Pier	1.27	1.27	1.57	1.57	
Bare land	8.07	6.42	14.61	13.17	
Road	6.65	7.65	9.12	10.71	
Green space	87.3	78.34	66.59	57.35	
Construction area	1.27	7.98	10.01	9.85	
Residential area	23.02	23.23	23.23	23.79	
Other facilities	3.22	6.29	6.32	7.65	
Container yard	2.23	5.33	5.49	10.91	

The areas of different land use types in the study area from 2009 to 2018 are shown in Table 2. There was a significant expansion in the northern part of the port area in 2012, relative to 2009 (Fig. 5A & 5B). Green areas, bare land, and water in the 2009 image were built on and expanded to become container yards, construction areas, roads, and other facilities in the later images. The area of these construction-related land use types increased by 6.71 hectares. Between 2012 and 2015, a 104-m-long, 23-m-wide, pier and ship-landing infrastructure was added to the northern port region, and the southern parts of the port area expanded (Fig. 5B & 5C). A total of 2.57 hectares originally classified as green space and construction area was converted into port land, with an associated 9.6-hectare increase in the area classified as bare land. Between 2015 and 2018, the port produces 2.81 hectares of land through occupied waters, mostly in the eastern parts of the study area, to provide a container yard and construction area. The original construction area in the western part of the port was converted to buildings that covered an area of 3.79 hectares. Container yards increased by 5.42 hectares compared to 2015.

Monitoring of changes to vegetation coverage

Five areas within the study area were randomly selected for accuracy verification using the 0.3-m-resolution high-definition image. The accuracy calculated for the five sample areas in 2009 was 82.5%, 88.1%, 94.6%, 91.3%, 87.5%, and the average accuracy was 88.8%. The accuracy of the five sample areas in 2018 was 81.2% 86.4%, 93.5%, 92.7%, and 87.2%, with an average accuracy of 88.2% (Fig. 6). These data were more than 80% accurate and meet the experimental requirements.

The fractional vegetation cover map of 5 km around the port was mapped (Fig. 7). In 2009, the boundary between the water and the vegetation is clearly defined (Fig. 7A). Within this region, 11.34% of the vegetation ranged from 0.2 to 0.6, 10.83% of vegetation ranged from 0.6 to 0.8, and 32.88% of the vegetation ranged from 0.8 to 1. A total of 43.71% of the vegetation, which covered an area of 10.94 km2, was between 0.6 and 1.

In 2018, 14.62% of the vegetation ranged from 0.2 to 0.6, 8.35% of the vegetation ranged from 0.6 to 0.8, and 33.9% of the vegetation ranged from 0.8 to 1 (Fig. 7B). The proportion of the vegetation between 0.8 and 1 was higher in 2018 (33.9%) than it was in 2009 (32.88%). A total of 44.25% of the vegetation, which covered an area of 10.58 km2, was between 0.6 and 1; these values are similar to those calculated for 2009 (Table 3). Vegetation coverage is an important indicator of the quality of the ecological environment in a region (Alo & Pontius, 2008; Xin, Xu & Zheng, 2008; Amarsaikhan et al., 2009; Chen & Wang, 2009; Zhang et al., 2010; Lu et al., 2017; Li et al., 2019). Thus, the comparison of the two images shows that the port expansion has not had a significant impact on the ecological status of the area, reflecting the higher overall greening level in the area.

Figure 6 Vegetation coverage accuracy verification map.

Five sets of samples of vegetation coverage accuracy verification map in 2009 (A&B, C&D, E&F, G&H, I&J), Five sets of samples of vegetation coverage accuracy verification map in 2018 (K&L, M&N, O&P, Q&R, S&T) (A, C, E, G, I): ©2009 Google Earth, (B, D, F, H, J): 2009 Landsat imagery courtesy of NASA Goddard Space Flight Center, (K, M, O, Q, S): ©2018 Google Earth, (L, N, P, R, T): 2018 Landsat imagery courtesy of NASA Goddard Space Flight Center.

Figure 7 Vegetation coverage map for 2009 (A) and 2018 (B).

Table 3 Statistics of vegetation coverage around the port of Ajmr in 2009 and 2018.

Years	Vegetation coverage	0–0.2	0.2–0.4	0.4–0.6	0.6–0.8	0.8–0.9	0.9–1	Total	
2009	area (km2)	11.25	1.25	1.59	2.71	3.52	4.71	25.03	
Proportion (%)	44.95	4.99	6.35	10.83	14.06	18.82	100.00	
2018	area (km2)	10.8	2.05	1.61	2.09	2.38	6.11	25.04	
Proportion (%)	43.13	8.19	6.43	8.35	9.50	24.40	100.00	

Analysis of changes in the port roads and buildings

The accuracy of road and building data on the revised 0.3-m-resolution high-definition image is better than 90%. During the nine-year expansion of the port, there were significant increases in the areas occupied by roads and buildings (Fig. 8). Seven new roads covering an area of 2.97 hectares appeared in the port; the new roads were mainly concentrated in the northeastern part of the port. A large number of container areas were added on the newly reclaimed land, so the number of transportation pathways for goods increased accordingly.

Figure 8 Port road and building change map for 2009 (A) and 2018 (B).

(A) ©2009 Google Earth, (B) ©2018 Google Earth.

Changes to buildings after the expansion included an increased number of buildings in the central and southern regions of the port. Expansion involved the addition of 22 new buildings with an area of 3.87 hectares. This large area of the port was converted at the expense of green space and water. The new roads and buildings accounted for 9.54% and 12.43% of the total area of the port, respectively, which is a relatively small proportion. The increase in the construction area reflects an increase in the population around the port, and the expansion of construction scale.

Port light change analysis

The port is adjacent to Panabo City to the north and Davao City to the south. Davao City is the only independent city in the Davao District of Northern Mindanao and is the most populous city, while the city of Panabo is a small city in the north of Davao. The time-series dynamic lighting monitoring of the 40 km ×40 km area around the port of Ajmr provided lighting data from 2009 to 2018 (Fig. 9). A comparison of the port lighting data with that of the adjacent cities showed that there was a significant increase in lighting at the port, especially in 2011 to 2015. During this time, the port’s lighting data was close to that of Davao, so grew significantly. The nine-year lighting dataset from the cities of Panabo and Davao also showed increases in lighting, but the rate of growth was slower than at the port.

Figure 9 Davao City, Panabo City and Ajmr Port light change curve in 2009–2018.

Differences between the 2009 and 2018 lighting images were calculated to obtain the nine-year lighting growth rate around the port (Fig. 10). The expansion of lighting from the northeast to the southwest is clear, highlighted by the red curve. From the high-definition image, it can be seen that the red curve passes over the coast southeast of Mindanao in the southern Philippines. The rate of growth of lighting along the coastline was commonly between 100 to 150, or greater. Panabo City and Davao City also had lighting growth rates between 100 and 150, and both are located on the coast. Before the expansion of Ajmr Port, the port’s economic development level was low. However, after nine years of continuous expansion, the rate of economic development in the region increased, indicating that construction at the port promoted local economic development. Ajmr Port, Panabo City, and Davao City form an economic belt on the southeast coast of Mindanao, which has promoted economic development on the coast.

Figure 10 Growth chart of the surrounding lights of the port in 2009–2018.

The lighting in large areas on the east side of the port grew at relatively low rates, between 0 and 50. The high-resolution images show that this slow-growth area is northwest of Davao Bay and Samal Island. The growth of lighting data in the sea areas is mainly due to ships. The large number of vessels in the water indicates the frequency of maritime transport trade in the region. The relevant data show that export volumes from Ajmr Port are large, and that there are numerous trading partners (Ahamad & Chua, 2013; Dhang & Partho, 2015; Zhai, Tang & Zhang, 2016). In 2015, the Sumifru Company began to export South American bananas to China. This company relies on China’s Dalian, Tianjin, Shanghai, Shenzhen, and other coastal ports for trade. It has established large processing, storage, and sales organizations in the major port cities of China that provide fresh fruit to the Chinese mainland. Relationships between China and the Philippines have benefited from this “21st Century Maritime Silk Road” (Zhai, Tang & Zhang, 2016). Through the continuous increase in sea-based trade, China has become the largest export market for bananas from the Philippines.

Conclusions

After nine years of expansion, the port area has gradually expanded from the central to the southern coastal area, with an increase of 21.68 hectares during the expansion period. After the expansion, the area of buildings and construction in the study area increased significantly, while the area of water and green areas decreased significantly, indicating that the port construction changed the land use structure of the area. From the perspective of vegetation coverage, the vegetation coverage within 5 km from the port is in good condition. After 9 years, the vegetation coverage in the region between 0.6 and 1 increased from 43.71% to 44.25%, reflecting the higher overall greening level in the region.

After 9 years of expansion, there are seven new roads in the northeastern part of the port, covering an area of 2.97 hectares. There are 22 new buildings in the central and southern ports, occupying an area of 3.87 hectares that is relatively large. By analyzing the increase in roads and buildings, it can be seen that the port’s comprehensive transportation capacity has improved, the population of the region has increased significantly. As the scale of construction has been continuously expanded, the prosperity as increased. By analyzing the changes in the light index, the light data from the northeast to the southwest in the region is very obvious, and it is clearly located along the coast, indicating that the economic development of the coastal zone is faster than other regions, and the coastal region has promoted the development of the inland region.

Discussion

The research that had been carried out was a unilateral study of the port. This paper has carried out remote sensing monitoring and analysis of the five aspects of economic development and ecological environment from the port of Ajmr in the Philippines from 2009 to 2018, which fills the gap in the comprehensive study of port construction monitoring. The small ports were monitored and analyzed, which was different from the traditional large port monitoring. It is found that the same problem exists in the construction of small ports: the problem between port construction and environmental protection.

This paper carries out monitoring and analysis from two aspects of ecological environment and economic development. The ecological environment refers to the general name of the quantity and quality of water resources, land resources, biological resources and climate resources that affect human survival and development, and is a complex ecosystem related to the sustainable development of society and economy. Because the ecological environment includes a wide range, research is not possible in every aspect. The research content of this paper monitors the ecological status of the area from two aspects. Remote sensing monitoring and analysis can also be carried out from the aspects of water resources and ecological resources. For example, Smith et al. (2007) used water quality, sediment quality, and benthic diversity data as baselines to assess the ecological status of coastal resources in the Gulf of Mexico. Cowell & Monk (1981) conducted a monitoring survey of the abundance of intertidal species near the dock at Valdez Port, Alaska.

The economy also includes many indicators, such as population, GDP, employment status, public policy and other statistics. This paper monitors and analyzes road and building changes and lighting growth. If you add some social and economic indicators to supplement, and use more targeted statistical analysis methods, you can get richer analysis results. For example, Kidd (2010) explores the impact of historical public policy and port expansion in Houston, Texas on the socioeconomic status of surrounding communities.

In this paper, the multi-temporal high spatial resolution remote sensing image is used to extract the elements by visual interpretation, monitoring the expansion of the port of Ajmr from 2009 to 2018. Other scholars have adopted methods such as comprehensive analysis, comprehensive index method, decision tree classification method, microscopic simulation model, and land use change index model. The method of establishing the model can be used for reference and can improve the accuracy of the prediction.

Based on the above research and analysis, the contradiction between the economic development of the port area and the local environmental protection still exists. How to reduce environmental damage and build an ecological port is still a focus of attention. This paper believes that: first, the environment can be protected by rational planning of land during the construction process. When occupying green space, it should add a green space in other places to keep the overall green area unchanged. Secondly, the environment can also be protected by improving the equipment inside the port, for example, using environmentally clean materials for loading and unloading goods and construction. Finally, effective degradation and treatment of wastewater and waste gas from port removal are also important measures to protect the environment.

Supplemental Information

Supplemental Information 1 2009 high-resolution imagery

Google Earth High Definition imagery

Click here for additional data file.

Supplemental Information 2 2012 high-resolution imagery

Google Earth High Definition imagery

Click here for additional data file.

Supplemental Information 3 2015 high-resolution imagery

Google Earth High Definition imagery

Click here for additional data file.

Supplemental Information 4 2018 high-resolution imagery

Google Earth High Definition imagery

Click here for additional data file.

Supplemental Information 5 2009 landsat image

Landsat imagery courtesy of NASA Goddard Space Flight Center and U.S. Geological Survey

Click here for additional data file.

Supplemental Information 6 2018 landsat image

Landsat imagery courtesy of NASA Goddard Space Flight Center and U.S. Geological Survey

Click here for additional data file.

The author is grateful to the State Key Laboratory of Remote Sensing Science of the Institute of Remote Sensing and Digital Earth of the Chinese Academy of Sciences for providing the data needed for this article. The author expresses her thanks to Professor Wu for his support, encouragement, and comments on a previous version of the paper. The author also expresses thanks for the methodological guidance given by Mr. Tian.

Additional Information and Declarations

Competing Interests

Author Contributions

Data Availability

The authors declare there are no competing interests.

Humei Li conceived and designed the experiments, performed the experiments, analyzed the data, prepared figures and/or tables, authored or reviewed drafts of the paper, approved the final draft.

Mingquan Wu conceived and designed the experiments.

Dinghui Tian and Lianxi Wu contributed reagents/materials/analysis tools.

Zheng Niu provide resources.

The following information was supplied regarding data availability:

Landsat images for 2009 and 2018 and high-definition images for 2009, 2012, 2015 and 2018 are available as Supplemental Files. Light data is available from the Earth Observation Group, NOAA National Centers for Environmental Information (NCEI) via available at https://www.ngdc.noaa.gov/eog/viirs/download_dnb_composites.html.

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
