# Peer review of "Monitoring and analysis of the expansion of the Ajmr Port, Davao City, Philippines using multi-source remote sensing data"

_PeerJ, doi:10.7717/peerj.7512_

## Round 0.1 · original submission · Major Revisions

The referees suggest that the submission may be publishable, but only after major revisions have been made to your manuscript. Therefore, I invite you to respond to their comments and revise your manuscript.

Reviewer 1 ·

Basic reporting

The paper didn't show enough details of the methodology. For example, the authors mentioned a model algorithm was established in Line 142 and the visual interpretation method was used in Line 149. However, no details of these methods were given in the paper.

Experimental design

The authors should improve the visualization of the experiment results, e.g. explicitly show the changing areas in Figure 2 and Figure 4.

Validity of the findings

The article fails to show the accuracies of land cover classification and change detection.

Additional comments

The authors should present the main contributions of this paper compared with existing studies in the section of Introduction.

·

Basic reporting

The article deals with the Monitoring and Analysis of a Small Port Expansion Based on Multi-source and Multi-temporal High-resolution Imagery. This is an interesting topic. However, the current title is not appropriate, I recommend the authors replace the Multi-temporal High-resolution Imagery with Remote sensing data. The writing is clear. Overall, the manuscript is publishable, but, needs a major revision first. There are many grammatical and structural mistakes in the manuscript and I am not able to highlight/revise all of them. So, please revise the manuscript and improve it considerably in terms of grammatical and structural mistakes.
The literature is suitable for the purpose of the article.

Experimental design

1) Within Scope of the PeerJ journal.
2) Research question well defined, relevant & meaningful. It is stated how the research fills an identified knowledge gap.
3) Structure conforms to technical & ethical standard.
4) Although the experimentation is appropriate, the paper would gain in interest if it would provide a more in-depth discussion about the selected topic.

Validity of the findings

1) Data is robust, statistically sound, & controlled.
2) Conclusions are well stated, linked to original research question & limited to supporting results.
3) Speculation is welcome.

Additional comments

1. The authors should thoroughly address the novelty of this study, because so many works have been done in this point.
2. The section of introduction did not well review and cite existed works.
3. The section of study area is unclear and redundant, there are many unrelated contents of your study.
4. Line 109 This paper studies the expansion process for the port of Ajmr, which has been under construction since 2002. However, nearly no expansion occurred from 2002 to 2009, so only images from 2009 to 2018 were selected for preprocessing. How can the authors prove this?
5. The spatial resolution of these selected remote sensing data are definitely different. The authors should address problem properly.
6. A sketch map should be added in the revision to illustrate the detailed processing.
7. The section of Survey methodology, I want to see some substantial details about the field validation.
8. The detailed symbols of different Land use should be added and well illustrated.
9. Line 158 the visual interpretation method was used to analyze the spatio-temporal dynamic changes in the land… What’s the accuracy of the detection? The accuracy assessment is essential to this study.
10. The equations should be listed in order.
11. I want to know why the 5% and 95% are regarded as the thresholds?
12. Currently, the results and analysis are separated. They should be an integration.
13. The discussion is missing and unsatisfied, the author should make the best effort to modify this section. I want to see a considerable improvement in this regard.
14. The conclusion I not well stated and is written more as an abstract. A clear conclusion needs to be written.
15. All tables/figures: Caption needs to completely explain the table, including what the abbreviations mean. Work on text clarification and formatting, improve figures, list of the abbreviations would be useful in this case to easily understand the text.

Reviewer 3 ·

Basic reporting

This manuscript failed to show why it is important to map small port expansion and did a poor job in demonstrating why this work is necessary. To improve, please focus on why mapping small port expansions are important, what are the current state-of-art methods to do so, what are the knowledge gaps, and how this work can contribute to fill in the gaps.

Experimental design

Research questions were no clear in this manuscript, and therefore I cannot evaluate whether the methods taken here is novel or helpful.
Plus, when mapping port expansion (Figure 2), only a very small area are used, but for vegetation cover change, the authors decided to use a much larger region. Are you assuming that a small port will have affect vegetation in a much larger region? If so, what is your basis for this assumption and why was this area in Figure 3 chosen?

Please clearly state your research questions, and also why your approach can answer these questions. And properly show your assumptions for port expansion and vegetation change.

Validity of the findings

No comments.

Additional comments

This manuscript might be OK for a course project, but not for a research journal. To work towards a manuscript worth publishing, I recommend the authors 1) clearly identify the importance of the research subject; 2) review the latest developments in your research area; 3) identify the current knowledge gap in this field (your research question); 4) state your proposed solutions and the outcomes.

---

## Round 0.2 · Major Revisions

The three referees suggest that the submission may be publishable, but only after some revisions have been made to your manuscript. Therefore, I invite you to respond to their comments (especially those comments from reviewer 3 who reported that their previous comments had not been addressed) and revise your manuscript accordingly.

Reviewer 1 ·

Basic reporting

The article has met the standards.

Experimental design

The article has met the standards.

Validity of the findings

The article has met the standards.

Additional comments

The authors have made suitable revision to my comments. Here are some minor points that should be revised before publication.
1. Please add a legend to Figure 6.
2. Please move the content of discussion to the Conclusion Section.
3. the accuracy is 1 m -> the resolution is 1 m.

·

Basic reporting

As I can see from the previous review, you have corrected most of the reviwers comments. Especially for the research questions, and also why your approach can answer these questions. I mean the scientific problems.

Experimental design

no

Validity of the findings

no

Additional comments

no

Reviewer 3 ·

Basic reporting

no comment

Experimental design

no comment

Validity of the findings

no comment

Additional comments

The authors did not address my previous comments and I have serious concerns about the quality of the paper. In addition to the previous comments, here are a few more.

1. The current discussion reads like a conclusion. Please do a research about what a decent discussion looks like and improve it.
2. L411 - I do not think the authors can conclude that the construction of the port did not have a significant impact on the ecology of the area. How do you define 'ecology'? Did you do any test?
3. L423-425 – these statements are not supported by your data. I am shocked to see these conclusions.

---

## Round 0.3 · accepted · Accept

It is a pleasure to inform you that the above manuscript has been finally accepted for publication in PeerJ.